# Expanding the tolerance of segmented Influenza A Virus genome using a balance compensation strategy

Xiujuan Zhao[1], Xiaojing Lin[2], Ping Li[2], Zinuo Chen[2], Chengcheng Zhang[2], Balaji Manicassamy[3], Lijun Rong[4]*, Qinghua Cui[1,5]*, Ruikun Du[1,5]*

1 Innovation Research Institute of Traditional Chinese Medicine, Shandong University of Traditional Chinese Medicine, Jinan, China, 2 College of Pharmacy, Shandong University of Traditional Chinese Medicine, Jinan, China, 3 Department of Microbiology and Immunology, University of Iowa, Iowa, United States of America, 4 Department of Microbiology and Immunology, College of Medicine, University of Illinois at Chicago, Chicago, United States of America, 5 Qingdao Academy of Chinese Medicinal Sciences, Shandong University of Traditional Chinese Medicine, Qingdao, China

* Lijun@uic.edu (LR); cuiqinghua1122@163.com (QC); duzi857@163.com (RD)

## Abstract

Reporter viruses provide powerful tools for both basic and applied virology studies, however, the creation and exploitation of reporter influenza A viruses (IAVs) have been hindered by the limited tolerance of the segmented genome to exogenous modifications. Interestingly, our previous study has demonstrated the underlying mechanism that foreign insertions reduce the replication/transcription capacity of the modified segment, impairing the delicate balance among the multiple segments during IAV infection. In the present study, we developed a "balance compensation" strategy by incorporating additional compensatory mutations during initial construction of recombinant IAVs to expand the tolerance of IAV genome. As a proof of concept, promoter-enhancing mutations were introduced within the modified segment to rectify the segments imbalance of a reporter influenza PR8-NS-Gluc virus, while directed optimization of the recombinant IAV was successfully achieved. Further, we generated recombinant IAVs expressing a much larger firefly luciferase (Fluc) by coupling with a much stronger compensatory enhancement, and established robust Fluc-based live-imaging mouse models of IAV infection. Our strategy feasibly expands the tolerance for foreign gene insertions in the segmented IAV genome, which opens up better opportunities to develop more versatile reporter IAVs as well as live attenuated influenza virus-based vaccines for other important human pathogens.

## Author summary

Foreign nucleotide insertions often interfere with the replication/transcription of the modified segment of IAV genome, impairing the delicate balance of the segmented genome during IAV infection. In general the larger the insertion is, the more the balance is impaired. The limited tolerance of IAV genome greatly restricts the creation and application of recombinant IAVs. In this study, we developed a "balance compensation"

**Data Availability Statement:** All relevant data are within the paper and its Supporting information files.

**Funding:** This work was supported by National Natural Science Foundation of China (82104134 to R.D.) and Key Technology Research and Development Program of Shandong, China (2020CXGC010505 to Q.C.). The funding body has no role in the study design, data collection and analysis, decision to publish, or preparation of the manuscript.

**Competing interests:** The authors have declared that no competing interests exist.

strategy to expand the tolerance of IAV genome. In coupled with appropriate compensatory enhancement during initial construction, recombinant IAVs harboring much larger foreign insertions are successfully generated and maintain genetically stable, facilitating their use as powerful tools, e.g., the reporter IAVs and live-attenuated influenza virus-vectored vaccines.

## Introduction

Influenza A virus (IAV) is a respiratory pathogen that causes seasonal epidemics and occasional pandemics globally [1]. Although there are several FDA-approved drugs available to treat IAV infections, including the viral ion channel M2 blockers (amantadine and rimantadine), the neuraminidase inhibitors (oseltamivir, zanamivir and peramivir) and a cap-dependent endonuclease inhibitor baloxavir marboxil [2,3], the emergence of drug resistance mutations illustrates the pressing need for novel anti-influenza therapeutics [4].

Recently, the rapid development of reverse genetics facilitates generation of replication-competent recombinant IAVs carrying varied reporter genes, allowing for rapid quantification of viral replication, and noninvasive imaging of infected tissues in living animals [5–9]. These live imaging animal models of IAV infection will have far-reaching benefits for novel antiviral developments. However, since the tolerance of IAV genome to foreign insertions is limited due to its segmented architecture, most of these reporter IAVs up to date are either attenuated or unstable during replication in cell culture or animals [5–9]. Moreover, the reporter genes are usually restricted to those with relatively small ones in size. To further optimize the reporter IAVs, a directed evolution strategy has been developed. For example, Katsura *et al.* and Cai *et al.* independently passaged reporter IAVs serially in mice, generating mouse-adapted variants with restored virulence and enhanced reporter gene expression [10,11]. Sequencing analysis revealed that mutations in the RNA-dependent RNA polymerase (RdRp) constituents contribute to the wildtype-like fitness in both cases, although the precise mutation sites differ [10,11]. Nonetheless, this directed evolution strategy is time-consuming and the desired outcome may not always be guaranteed. Thus it is desirable to develop a general method that allows free manipulation of IAV genome.

The genome of IAV consists of eight negative-stranded viral RNA (vRNA) segments, of which each segment encodes one or two major open reading frames (ORFs), including PB2 (Polymerase basic 2), PB1 (Polymerase basic 1), PA (Polymerase acid), HA (Hemagglutinin), NP (Nucleoprotein), NA (Neuraminidase), M (Matrix proteins) and NS (Non-structural proteins). The central coding region of individual vRNAs is flanked by conserved 3' and 5' non-coding terminal sequences [12]. The first 13 nucleotides (nts) at the 5' end and the first 12 nts at the 3' end of vRNAs that are partially complementary interact to form promoter structures for viral RdRp complex [13,14].

Both the conformations of influenza polymerase complex and promoter structure are flexible and interactively regulate the dynamical initiation of transcription and replication [14,15]. In addition, the multiple segments compete with each other to recognize viral polymerase for replication and transcription [16]. For instance, if the amount of RdRp complexes is limited, the replication/transcription of one segment vRNA would be negatively affected by the presence of the other seven counterparts, but this competition could be relieved as polymerase subunits accumulates [16]. This is consistent with the finding that the replication and transcription of the different segments follow different kinetics during IAV infection, reflecting a delicate dynamical balance of the segmented genome [17].

The balance of the multiple segments during IAV infection is vulnerable to artificial modifications. For example, it was reported that specific mutations in the promoters can enhance the levels of vRNA and mRNA [18,19], however, upon incorporation of these promoter-upregulating mutations, the recombinant IAV attenuates in replication, highlighting the critical need for the delicate balance between replication, transcription and protein expression of IAV segments [19]. In addition, many other factors including the segment-specific non-coding regions (NCRs), the length of coding regions, as well as the inherent activity and template preference of viral RdRp are involved in the competition [16,17,20].

In our previous report, we demonstrated that foreign insertions can drastically reduce the replication/transcription capacity of the modified segment due to inevitable increase in segment length, adversely impacting the balance of the multiple segments in the levels of vRNA and mRNA during IAV infection [21]. For example, when the imbalance occurred to the NS segment: First, inadequate vRNAs of modified NS are available for incorporation into progeny virions, producing a large amount of NS-null noninfectious particles [21]; Second, the expression of NS1 proteins decreased accordingly and may no longer counteract the host immune responses efficiently [22]; Third, the accumulation of NEPs which play a pivotal role to mediate nuclear export of viral ribonucleoproteins was greatly affected [23,24]. Together these mechanisms lead to a reduced replication and attenuation of the reporter IAVs.

In this study, we propose that incorporation of compensatory enhancing mutations during initial construction of a reporter IAV can rebalance the multiple segments, and subsequently reduce or eliminate the mechanism of attenuation. As a proof of concept, a reporter influenza PR8-NS-Gluc virus which carries a *Gaussia* luciferase (Gluc) gene fusing to NS1 was used as a starting point [25], and the aforementioned enhancing mutations were introduced into the promoter elements of the modified NS-Gluc segment to rectify the imbalance. We show that an optimized reporter IAV that shows improved replication kinetics and virulence was generated. Moreover, the tolerance of IAV genome can be further expanded using this "balance compensation" strategy. In conjugation with stronger compensatory enhancements, a recombinant IAV carrying a much larger firefly luciferase (Fluc) gene was subsequently generated.

## Results

### Evaluation of replication-enhancing mutations of IAV vRNA segments

Diverse structures could be formed from the 3'- and 5'-terminal nucleotides of influenza vRNA segments, including cockscrew, panhandle, fork and hook-like structures, dynamically regulating the transcription and replication of vRNA during virus life cycle [26]. Interestingly, it has been well documented that the panhandle structure is involved in the initiation of replication/transcription, while panhandle-stabilizing mutations within the conserved 3'-NCR can enhance the promoter activity [16,18]. These panhandle stabilizing mutations have potential to be employed as compensatory enhancements for reporter IAVs.

We first evaluated how panhandle-stabilizing mutations enhanced vRNA replication, transcription and subsequent protein expression, especially in competition with its natural counterpart. To this end, a dual-template reporter RdRp assay was employed. In brief, the firefly and renilla luciferase (Fluc and Rluc) genes were flanked by 3'- and 5'- NCRs of the NS segment and inserted in a negative sense orientation under the control of the human RNA polymerase I (Pol-I) promoter, generating IAV minigene expressing plasmids pNCR$_{NS}$-Fluc and pNCR$_{NS}$-Rluc, respectively. Mutations of interest were then introduced into pNCR$_{NS}$-Fluc, producing various mutants (pNCRx-Fluc), while pNCR$_{NS}$-Rluc was set as a constant competitive control. When pNCRx-Fluc and pNCR$_{NS}$-Rluc were co-transfected into 293T cells expressing IAV RdRp components and NP, mini vRNA reporters NCRx-Fluc and NCR$_{NS}$-

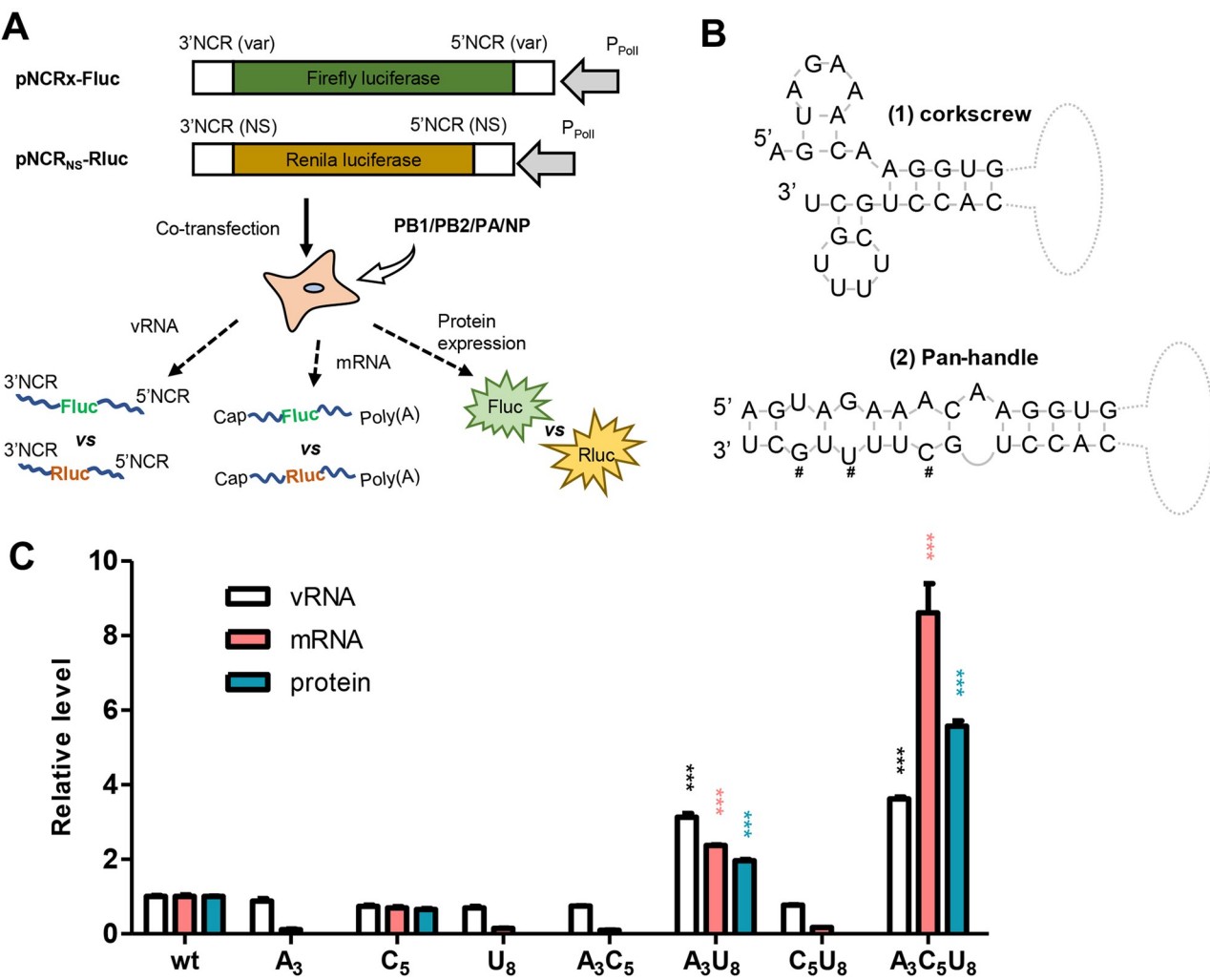

**Fig 1. Pan-handle stabilizing mutations at the 3'-NCR enhances the transcription/replication of influenza vRNA.** (A) A schematic overview of the dual-template reporter constructs and the cell-based dual-template RdRp assay. (B) Presentation of two of the proposed vRNA promoter structure models. #, indicates the pan-handle stabilizing mutation site. (C) Normalized ratio of firefly to Renilla luciferase (Fluc/Rluc) at the vRNA, mRNA as well as protein expression levels. ***, $p < 0.001$, students' t test.

Rluc were expected to compete with each other to bind RdRp complex. We thus measured the relative expression of vRNA and mRNA from the dual-templates by qRT-PCR as a measure of the replication and transcription efficacy of mutated versus wildtype vRNA promoter. Fig 1A shows a schematic overview of the dual luciferase reporter-based competition assay.

As described previously, three pan-handle stabilizing mutations at 3'-NCR (G3A, U5C, C8U) and the varied combinations were introduced into NCR_NS-Fluc separately by PCR using primers listed in S1 Table, and dual competition assay was conducted using minigenome NCR_NS-Rluc as a competitive control (Fig 1B) [16,18]. As shown in Fig 1C, double mutations G3A/C8U could significantly enhance both vRNA replication and transcription, while triple mutations G3A/U5C/C8U exhibited stronger enhancement, providing two sets of potent compensatory enhancement (CE). Here we will refer the G3A/C8U and G3A/U5C/C8U mutations as CE1 and CE2, respectively.

It is worth noting that mutations G3A, C8U, G3A/U5C, U5C/C8U drastically impaired the transcription and subsequent protein expression without affecting vRNA replication,

indicating that paired G3/C8 or A3/U8 is essential for the formation of transcription favored structures, e.g., the corkscrew like structure (Fig 1B).

## Compensatory promoter enhancements can rebalance vRNA segments of reporter IAV and restore the wildtype-like fitness *in-vitro* and *in-vivo*

To determine whether the compensatory enhancements can restore the replication kinetics and virulence of the reporter influenza PR8-NS-Gluc virus, the two sets of promoter-upregulating mutations were incorporated into the modified NS-Gluc segment (Fig 2A) [25]. Through a modified dual-template RdRp assay with wildtype M segment as a competitive control template, the expression of the mutated NS-Gluc vRNA was compared to the original NS-Gluc and natural NS by qRT-PCR analysis. As anticipated, both sets of mutations CE1 and CE2 significantly increased the replication of NS-Gluc vRNA (Fig 2B). Moreover, it is noteworthy that segment NS^CE1-Gluc which carries G3A/C8U mutations could replicate similarly in efficacy to natural NS, while segment NS^CE2-Gluc carrying G3A/U5C/C8U mutations replicated more efficiently. Similar compensatory enhancement was observed for vRNA transcription (S1 Fig).

Next, reporter influenza PR8-NS-Gluc viruses harboring additional CE1 or CE2 mutations at 3'-NCR of the modified NS-Gluc segment were rescued using reverse genetics system and were designated as PR8-NS^CE1-Gluc and PR8-NS^CE2-Gluc, respectively. Interestingly, the PR8-NS^CE1-Gluc virus showed improved replication kinetics as compared to original

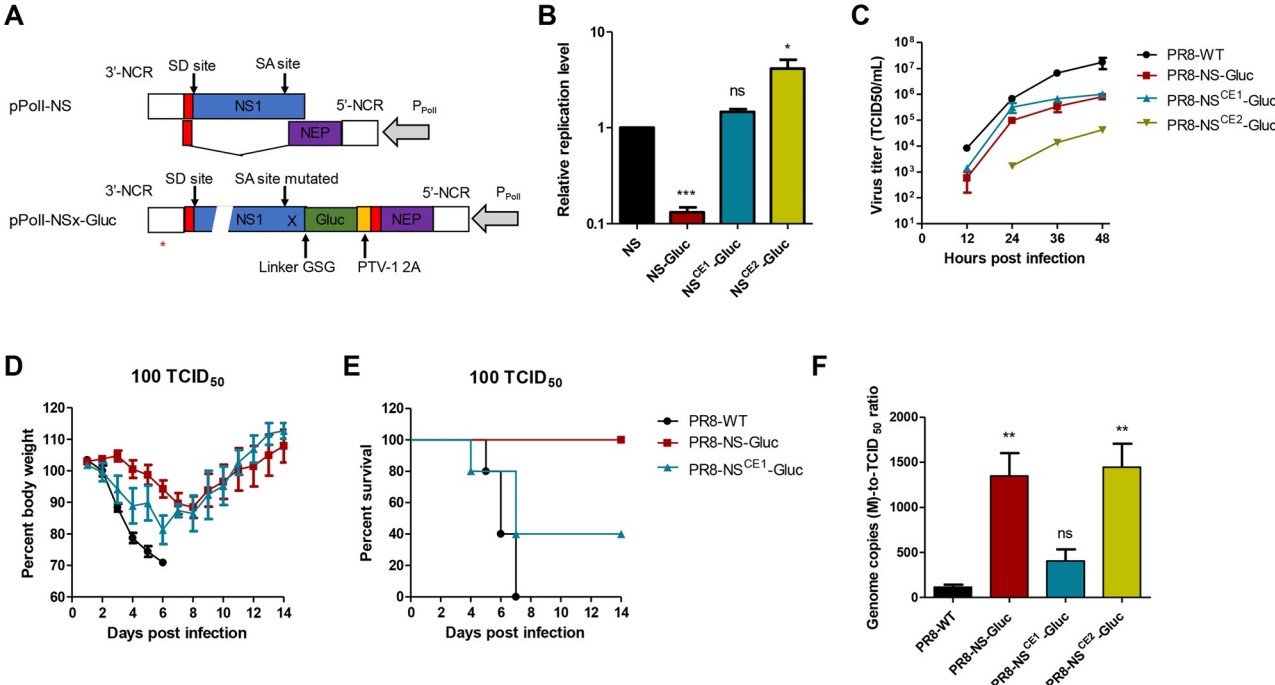

**Fig 2. Compensatory enhancement restores the replication kinetics and virulence of a reporter influenza PR8-NS-Gluc virus.** (A) Construction of modified NS-Gluc segment. *, replication enhancing mutations. (B) Relative replication efficacies of NS-derived vRNAs normalized to M. (C) *In vitro* replication kinetics of wildtype and modified influenza viruses. (D and E) Female BALB/c mice were infected with 100 TCID$_{50}$ of indicated viruses, and the body weight (D) and survival (E) were monitored for 14 days. (F) Relative genome copies to virus titer ratio of the parental PR8-WT and recombinant IAVs PR8-NS-Gluc, PR8-NS^CE1-Gluc and PR8- NS^CE2-Gluc viruses. Indicated viruses derived from MDCK were titrated and extracted for vRNAs followed by quantification of segment M by qPCR. The ratio of genome copies to virus titer were then evaluated. The standard deviations were calculated based on three replicates. *, $p<0.05$; ***, $p<0.001$; ns, no significance; students' *t* test.

PR8-NS-Gluc, although decreased replication was still observed compared to the parental wildtype PR8 virus (PR8-WT); however, PR8-NS$^{CE2}$-Gluc replicated even less efficiently than the original PR8-NS-Gluc virus (Fig 2C). Together, these results suggest that proper compensation to the modified segment can restore the wildtype-like *in-vitro* replication kinetics of reporter IAVs.

To determine whether CE1 could rescue the attenuated virulence of reporter PR8-NS-Gluc virus, the lethality of PR8-NS$^{CE1}$-Gluc to mice was assessed alongside with PR8-WT and PR8-NS-Gluc viruses as controls. At a dose of 100 TCID$_{50}$, PR8-WT infection caused rapid body weight loss of mice, while delayed and mild body weight loss were observed for PR8-NS-Gluc infected mice, indicating drastic attenuation of PR8-NS-Gluc virus (Fig 2D). Interestingly, PR8-NS$^{CE1}$-Gluc virus induced more rapid and severe body weight loss compared to the original PR8-NS-Gluc virus, although it was still not as virulent as the parental PR8-WT virus (Fig 2D). Similar enhancement could also be reflected by the lethality data, as PR8-NS-Gluc virus was sublethal at indicated dose, PR8-NS$^{CE1}$-Gluc caused 60% lethality (Fig 2E). These data clearly demonstrate that the *in vivo* fitness of reporter IAVs can be, at least partially, restored by rebalancing the levels of modified genome segments.

Our previous study has demonstrated that the ratio of genome copies to infective virus of PR8-NS-Gluc is much higher than the parental PR8-WT virus [21], suggesting an increased production of defective interfering (DI) particles. We therefore assessed the effect of CE incorporation on DI particles production. Consistently, the genome copies to infective virus ratio of PR8-NS-Gluc was about 11.6-folds compared to PR8-WT, while the PR8-NS$^{CE1}$-Gluc virus showed a 2.5-folds increase only, which was not statistically significant (Fig 2F). These results suggest that CE1 incorporation can reduce the abnormal production of DI particles caused by foreign gene insertion. However, the PR8-NS$^{CE2}$-Gluc virus produced 11.4-folds more DI particles than PR8-WT virus (Fig 2F), which is in accordance with the overcompensation of CE2 incorporation to genome balance and the attenuated replication of PR8-NS$^{CE2}$-Gluc virus (Fig 2B and 2C).

## Compensatory enhancement augments reporter protein expression

Considering CE1 could restore the replication capacity of the modified segment NS-Gluc, we reasoned that it was likely that the reporter protein expression could also increase correspondingly. To validate this, the kinetics of Gluc expression during PR8-NS$^{CE1}$-Gluc replication was monitored (Fig 3A). Further, the relative reporter protein expression level was determined by normalizing the bioluminescence signal to viral titer. As expected, Gluc expression in the PR8-NS$^{CE1}$-Gluc infected cells was significantly augmented by more than 10-folds (Fig 3B).

The reporter influenza PR8-NS$^{CE1}$-Gluc was next adapted for high-throughput antiviral screening. The signal-to-noise (S/N) ratio with PR8-NS$^{CE1}$-Gluc was boosted from 24 to 1109 compared to the original PR8-NS-Gluc based screening, while the coefficient of variation (CV) and Z' score were not affected (Fig 3C). Considering S/N ratio is one of the most important factors of quality control for high-throughput screens, and a higher S/N value usually means higher sensitivity, the optimized PR8-NS$^{CE1}$-Gluc virus described here can be used as a powerful platform for antiviral screening.

*In-vivo* bioluminescence imaging (BLI) of viral infection provides a valuable tool to investigate virus-host interactions and evaluate antiviral therapies. Our previous study revealed that PR8-NS-Gluc infected mice failed for *in-vivo* imaging, likely due to the combined effects of low expression level of Gluc and low transmission light generated by Gluc through tissues. Since PR8-NS$^{CE1}$-Gluc exhibited enhanced virulence and augmented Gluc expression compared to the original PR8-NS-Gluc, we asked whether PR8-NS$^{CE1}$-Gluc virus could be used for

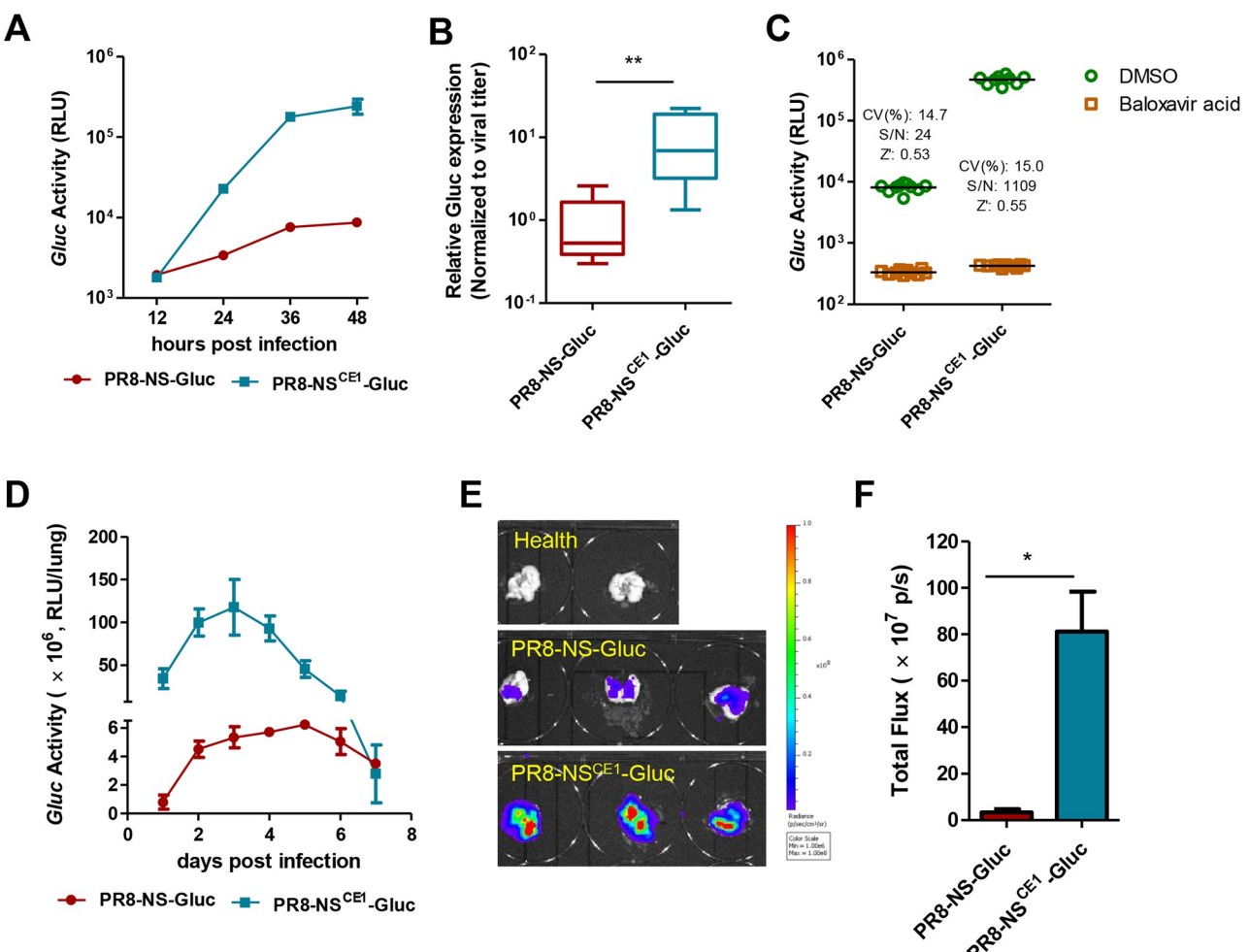

**Fig 3. Compensatory enhancement augments the expression of reporter genes.** (A) Kinetics of reporter luciferase expression during virus replication *in vitro*. (B) Normalized Gluc activity to virus titers of identical time points (relative Gluc activity/viral titer) for original PR8-NS-Gluc and mutated PR8-NS$^{CE1}$-Gluc viruses. (C) Adaptation of the reporter viruses as an *in vitro* high-throughput screening approach. CV, coefficient of variation; S/N, signal-to-noise ratio. (D) Kinetics of reporter luciferase expression during virus replication in mice. (E) *Ex-vivo* imaging of excised mice lungs. (F) The signals of *ex-vivo* imaging derived from mice lungs infected by reporter PR8-NS-Gluc and PR8-NS$^{CE1}$-Gluc viruses.

*in-vivo* imaging in mice. To test this, female BALB/c mice (4 weeks old) were infected with 100 TCID$_{50}$ of influenza PR8-NS$^{CE1}$-Gluc and PR8-NS-Gluc separately, and the kinetics of Gluc expression in the lung tissues were monitored. As shown in Fig 3D, although similar trend of rising and declining of Gluc expression were observed in both groups, which reflected similar kinetics of virus spreading and clearance, the bioluminescence level in lung tissues from the PR8-NS$^{CE1}$-Gluc infected mice was roughly 20-fold higher than those from the PR8-NS-Gluc infected mice throughout the course of infection. Moreover, *ex-vivo* imaging revealed similar bioluminescence enhancement of PR8-NS$^{CE1}$-Gluc compared to PR8-NS-Gluc (Fig 3E and 3F). However, live imaging in the PR8-NS$^{CE1}$-Gluc infected mice was not successful after several rounds of attempt.

## Development of a recombinant IAV carrying a large firefly luciferase gene

Considering our success in directed optimization of the reporter influenza PR8-NS-Gluc virus, we asked whether the "balance compensation" strategy would allow IAV genome to tolerate

larger foreign insertions, e.g., the Fluc gene. Fluc possesses unique advantages in live imaging, including the relative ease of substrate administration (intraperitoneal injection), longer duration of signal, low cost, and more importantly, the longer peak emission wavelength (~612 nm), which contributes to its high transmission ability through tissues, providing a high level of sensitivity. However, since Fluc is large in size (~1,650 bp) and the IAV genome might not tolerate such a large insertion.

In this study, we replaced the Gluc gene of the modified NS-Gluc segment with Fluc, generating NS-Fluc. An additional 2A protease sequence was inserted between NS1 and Fluc to avoid potential interfere of Fluc, due to its larger size, to the function of NS1 (Fig 4A) [27]. In addition, NS-Fluc constructs incorporating CE1 or CE2 mutations at 3'-NCR were also generated, designated as NS$^{CE1}$-Fluc and NS$^{CE2}$-Fluc, respectively. The replication capacities of the NS-Fluc and two mutants were determined using dual-template RdRp assay. As shown in Fig 4B, the replication of NS-Fluc was greatly impaired, with a signal roughly 0.2% of the parental NS, which was even worse than NS-Gluc (10% of the parental NS). In contrast, both CE1 and CE2 mutations at 3'-NCR improved the replication efficacy of vRNA NS-Fluc, by 3.8 and 9.4- fold, respectively.

Next, we tried to incorporate the original NS-Fluc and two mutants into recombinant IAVs by reverse genetics. Not surprisingly, no recombinant IAV harboring the original NS-Fluc segment was generated, because the replication capacity of NS-Fluc segment was severely compromised. In contrast, both recombinant IAVs harboring NS$^{CE1}$-Fluc or NS$^{CE2}$-Fluc were rescued successfully. While the NS$^{CE1}$-Fluc segment appeared to be unstable since the signal of exogenous reporter was lost rapidly within three passages when propagated in chicken embryo, the PR8-NS$^{CE2}$-Fluc virus exhibited good stability for at least five passages (Fig 4C and S2 Fig). Interestingly, deep sequencing of the fifth passage of PR8-NS$^{CE2}$-Fluc virus identified a U1G mutation with 20% frequency at the 3'-NCR of NA segment (S3 Fig).

As shown in Fig 4D, PR8-NS$^{CE2}$-Fluc replicated efficiently in MDCK cells, although the peak titer was 1-log lower than the parental PR8-WT virus. In a mouse model, PR8-NS$^{CE2}$-Fluc was shown to be infectious and pathogenic, causing rapid body weight loss and leading to 100% lethality at high challenge doses (Fig 4E).

## Establishment of a robust live imaging animal model of IAV infection

We next evaluated the potential use of PR8-NS$^{CE2}$-Fluc in a live imaging mouse model of IAV infection. A group of mice were inoculated intranasally with PR8-NS$^{CE2}$-Fluc at a sub-lethal dose (1,000 TCID$_{50}$), and the real-time bioluminescence was performed every day for nine days to monitor temporal changes in virus replication and tissue distribution over time (Fig 5A). It could be observed that viral infection was initiated in nasal-related tissues and both lobes of the lungs, as previously described [6,9,28]. The dynamics of IAV infection was also clearly delineated. As shown in Fig 5B, the BLI signal could be detected as early as day 1 post infection (p.i.), then elevated and peaks at day 2 or 3 p.i., while the signal decreased greatly after 6 days p.i., indicating virus clearance. In a separate experiment, cohorts of mice were imaged and necropsied at 1, 3, 5, 7, 9 days p.i. (3 mice per time point), and the high correlation between BLI signal intensity and virus load in lungs was observed ($r^2 = 0.68$, $p < 0.001$; Fig 5C), indicating that BLI measure can accurately reflect the virus replication *in vivo*.

To further test if our BLI-based mouse model of IAV infection could be used to evaluate efficacy of antiviral therapeutics, the mice were administrated with 10 or 30 mg/kg/day oseltamivir phosphate per oral twice a day starting at 2 hours prior to virus challenge. Mice treated with vehicles only were used as controls. BLI was acquired at day 2 and 5 p.i.. Both doses of oseltamivir reduced the signal intensities greatly (Fig 5D and 5E). These results demonstrate

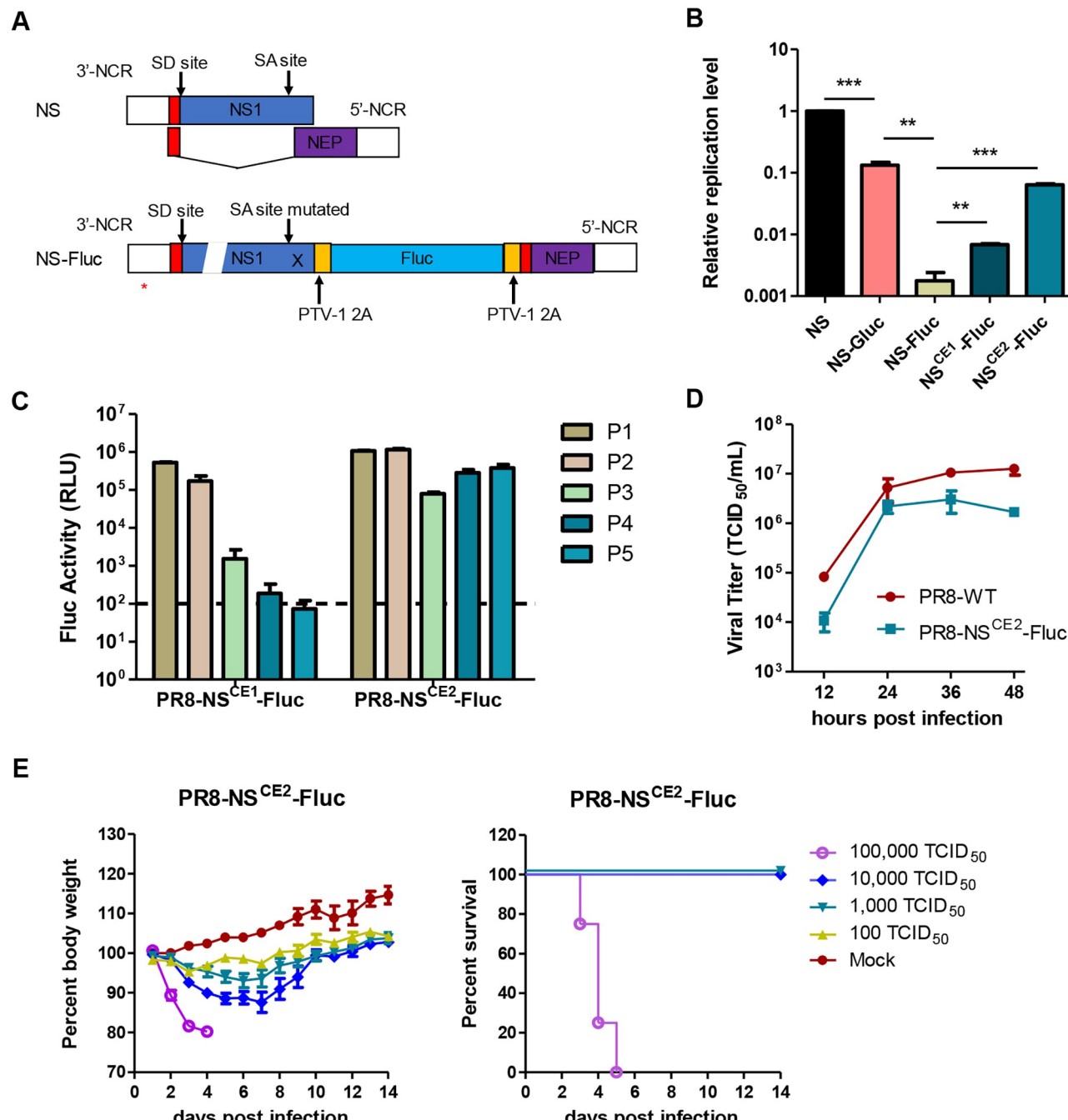

**Fig 4. Generation and characterization of a recombinant IAV expressing Fluc gene.** (A) Schematic representation of the natural NS segment and modified NS encoding the Fluc reporter gene. SD/SA—Splice donor/acceptor sites. (B) Relative replication efficacies of NS-derived vRNAs normalized to M. (C) MDCK cells were infected with PR8-NS^CE1-Fluc or PR8-NS^CE2-FLuc viruses from serial passage experiments in eggs (passages 1 to 5) at an MOI of 0.01. At 24 h.p.i. cells were harvested, and luciferase assays were performed. (D) Comparison of the *in vitro* replication kinetics of PR8-WT and reporter PR8-NS^CE2-Fluc viruses. (E) PR8-NS^CE2-Fluc infection induces severe body weight loss and lethal disease in mice.

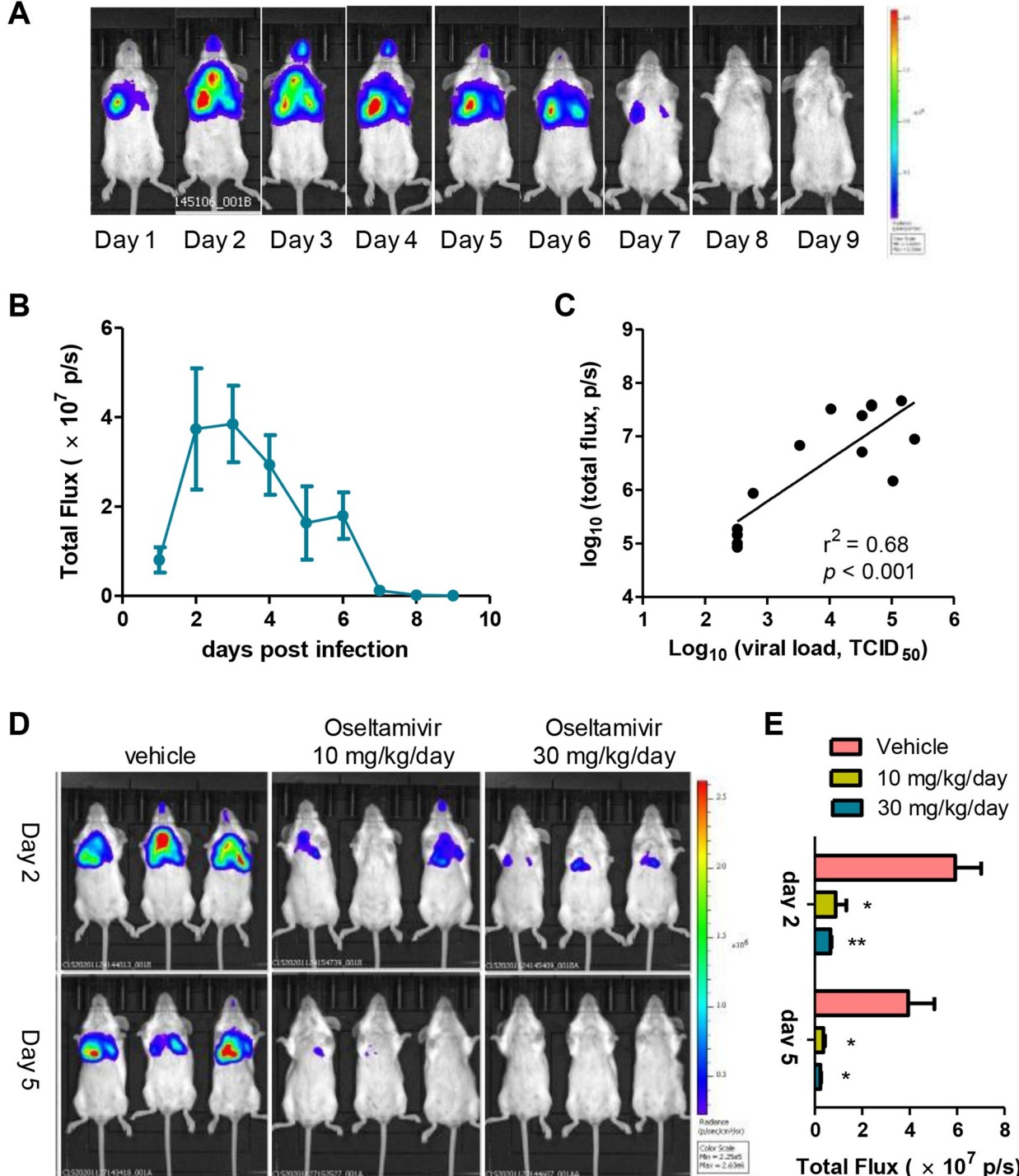

**Fig 5. Establishment of an *in vivo* imaging mouse model of IAV infection.** (A) Mice were infected with 1,000 $TCID_{50}$ of $PR8-NS^{CE2}$-Fluc virus and the bioluminescence was monitored daily for 9 days. (B) Bioluminescence imaging data corresponding to the time course of $PR8-NS^{CE2}$-Fluc infection. (C) In a separate experiment, *in vivo* imaging was performed and mice were then euthanized for determination of viral load in lungs. The bioluminescence density and viral load from individual animals were plotted against each other from time courses of $PR8-NS^{CE2}$-Fluc infection. (D) $PR8-NS^{CE2}$-Fluc infected mice were treated with 10–30 mg/kg/day of oseltamivir phosphate or vehicle only, and the *in vivo* imaging was carried out on days 2 and 5. (E) Data correspond to the *in vivo* imaging. *, $p<0.05$; **, $p<0.01$; students' t test.

that our PR8-NS$^{CE2}$-Fluc based live-imaging mouse model accurately reflects the *in vivo* efficacy of oseltamivir.

## Development of a live imaging animal model of H3N2 IAV infection

To investigate whether we could develop a similar live imaging mouse model with another IAV strain, we developed and evaluated an A/X31(H3N2)-based live imaging strain. The X31 virus is a reassortant virus harboring six internal segments from A/PR8 as backbone and the HA and NA segments from a mouse adapted virus of the H3N2 Hong Kong epidemic strain, and it has been broadly used to establish mouse models of H3N2 infection [29].

First, we generated a reporter X31-NS$^{CE2}$-Fluc virus containing the six internal segments from the PR8-NS$^{CE2}$-Fluc as backbone by reverse genetics. Subsequently, the X31-NS$^{CE2}$-Fluc virus was characterized both *in vitro* and *in vivo*. Predictably, the *in vitro* replication of X31-NS$^{CE2}$-Fluc was reduced compared to wildtype X31 virus, with peak virus tier of more than 2-log lower (Fig 6A). In addition, the X31-NS$^{CE2}$-Fluc virus was attenuated in mice compared to the X31 virus (Fig 6B). Nonetheless, the X31-NS$^{CE2}$-Fluc virus exhibited similar lethality as the recombinant PR8-NS$^{CE2}$-Fluc virus.

The live imaging of X31-NS$^{CE2}$-Fluc in mice was evaluated. BALB/c mice were grouped and inoculated intranasally with X31-NS$^{CE2}$-Fluc at a series of doses from $10^2$ to $10^5$ TCID$_{50}$, followed by monitoring real-time bioluminescence at day 1, 2, 3, 5 and 8 post challenge (Fig 6C). As shown in Fig 6D, the kinetics of BLI signal clearly showed the initiation, spreading

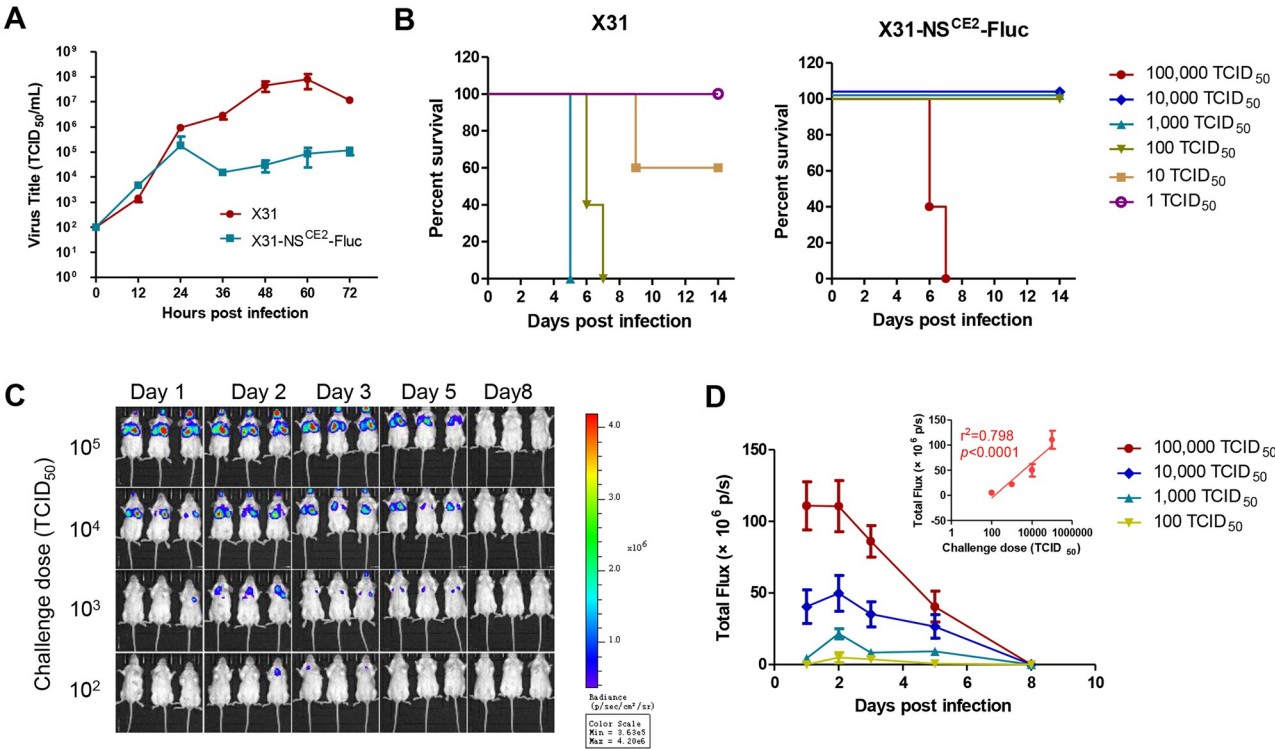

**Fig 6. Establishment of an *in vivo* imaging mouse model of IAV subtype H3N2 infection.** (A) Comparison of the *in vitro* replication kinetics of wildtype X31 and reporter X31-NS$^{CE2}$-Fluc viruses. (B) The lethality of wildtype X31 and reporter X31-NS$^{CE2}$-Fluc viruses in mouse models. (C) Mice were infected with series doses of X31-NS$^{CE2}$-Fluc virus and the bioluminescence was monitored at days 1, 2, 3, 5, and 8 post challenge. (D) The kinetics of bioluminescence imaging in mice infected with varies doses of X31-NS$^{CE2}$-Fluc virus. Inset, linearization analysis of the bioluminescence signals at day 2 post challenge to challenge doses.

and clearance of virus infections. Moreover, the signals at day 2 from different groups correlated well to the challenge doses (Fig 6D), suggesting this IAV strain can be adapted to be used for live imaging.

## Discussion

The limited tolerance of IAV genome to foreign gene insertion has hampered the development of recombinant IAVs as tools, e.g, reporter viruses [5,6,30], and live-attenuated influenza virus-vectored vaccines bearing foreign antigens [31–33]. Previously, we have unraveled an underlying mechanism that foreign insertions may cause reduced transcription/replication capacity of the modified vRNA segment in competition with other wildtype segments, impairing the balance of the segmented genome during infection. Thus, the balanced profiles of the eight segments at the levels of vRNA, mRNA and protein expression are all adversely affected, leading to defective genome packaging into progeny virions [21]. As a consequence, the replication and virulence of recombinant IAVs are attenuated, and the foreign insertions may be lost rapidly during virus passaging. In the present study, we developed a "balance compensation" strategy to expand the tolerance of the segmented IAV genome to facilitate genome manipulations. As depicted in Fig 7, directed optimization of recombinant IAVs is achieved by incorporating compensatory enhancements (CEs) to rectify the reduced replication of modified segment and genome imbalance.

In this study, a well-studied set of replication-enhancing mutations CE1 at the promoter element was introduced into the modified NS-Gluc segment of the reporter influenza PR8-NS-Gluc virus, and it was shown that CE1 could correct the reduced replication/transcription of modified NS-Gluc segment. Further, CE1 incorporation not only restored the wildtype-like fitness of the reporter virus (see Fig 2), but also significantly enhanced the reporter Gluc expression (see Fig 3). Importantly, we successfully generated stable replication-competent recombinant IAVs integrating a much larger Fluc gene coupled with a much stronger set of replication-enhancing mutations CE2 (see Figs 4 and 6).

Our work demonstrates there is delicate balance of the segmented IAV genome for genome manipulations. For instance, since CE1 can compensate the reduced replication/transcription efficacy of NS-Gluc to the level of natural NS, and CE2 leads to overcompensation (see Fig 2B), it seems reasonable to assume that CE1 but not CE2 was more suitable for directed optimization of the recombinant PR8-NS-Gluc virus. However, since Fluc is much larger and its insertion to the genome could impair the balance of segmented genome more drastically, the CE2 but not CE1 was more optimal for generating a more stable recombinant PR8-NS-Fluc virus (see Fig 4). It is noteworthy that CE2 is likely not optimal enough to compensate the reduced replication of NS-Fluc segment, and additional compensation(s) is likely required to generate more stable IAVs (see Fig 4B). In addition, since PR8 strain was laboratory-adapted and likely harbors increased tolerance for genome rearrangement, generation of recombinant IAVs from clinical isolates may require more optimization for genome balance.

Our "balance compensation" strategy, described here, is also segment dependent. Besides NS, other segments including PB1, PB2, PA, NA and NP have also been engineered to create reporter IAVs [5–7,30,34]. Considering the promoter sequences is highly conserved among all segments, the aforementioned promoter enhancing mutations should also apply to optimize these reporter viruses. In addition, many segment-specific enhancement candidates should be considered. First, as a U/C polymorphism at position 4 exists within the 3'-NCR of IAV segments, U4 contributes to a higher transcription/replication capacity compared to C4 [35]. For those segments that carry original C4 at the 3'-NCR, a C4 to U4 mutation should be employed as compensatory enhancement. Second, the segment-specific NCR sequences are also involved

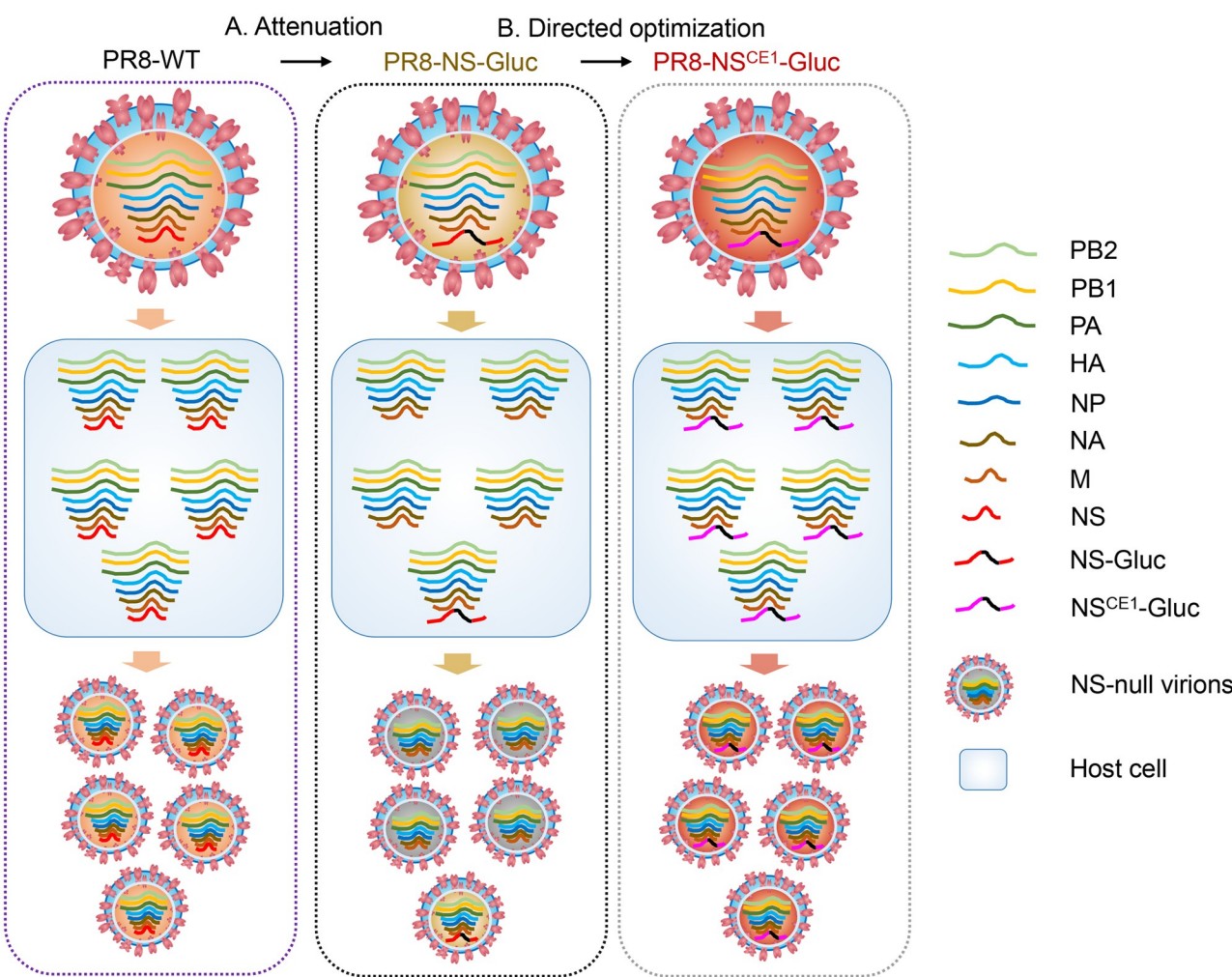

**Fig 7. Representative diagram of the molecular mechanism underlying attenuation of reporter IAVs and the proposed "balance compensation" strategy for directed optimization.** (A) The NS segment was modified with foreign insertions, resulting in reduced replication/transcription of the NS-derived vRNA and impaired balance of the multiple segments. Subsequently, a large proportion of progeny virions are NS-null and non-infectious. (B) For directed optimization of the reporter IAV, proper compensatory enhancement was incorporated during initial construction. The replication/transcription of NS-derived vRNA was specifically increased, while rebalance of the segmented genome could be achieved, restoring the wildtype-like fitness. Of note, the multiple segments in wildtype PR8 virus infected cells were shown in equal molar ratio for conceptual illustration only.

in vRNA transcription/replication [36–38]. For example, it was reported that a U13 to C13 mutation in the 3′ end of the NA gene promoted the expression of viral RNA and protein, while mutation of other sites within the UTR could differentially regulate viral genomic transcription and translation [36]. We speculate that a U13 to C13 mutation at the variable NCR sequence of the modified NA would compensate to the attenuation [7]. Third, since the accumulation of vRNA and mRNA during IAV infection is dynamic and segment-specific, both the inherent activity and template preference of viral RdRp may be involved in the regulation [17,20]. As mentioned above, Katsura *et al.* generated a mouse-adapted reporter influenza Venus-PR8 virus carrying a Venus gene within NS segment, and identified a PB2-E712D substitution that could stabilize the foreign gene insertion and restore wildtype-like replicative ability and virulence in mice [10,22]. Mechanistic studies revealed that the polymerase fidelity was not affected by PB2-E712D substitution [22], and the inherent polymerase activity of

PB2-E712D is even lower than that of wildtype PB2 [10]. Notably, considering the inherent polymerase activity was determined using an NP-derived template [10], and it was demonstrated that PB2-E712D enhanced the transcription/replication efficiency of the modified NS as compared to NP [22], we speculate that the mutated polymerase may possess an increased preference to NS segment over NP. Further studies are needed to investigate the preference profile of the mutated polymerase to the eight segments. It is possible that these enhancement candidates can be used independently or in combination to achieve appropriate compensation, i.e., to rebalance the segmented genome.

In summary, we have developed a "balance compensation" strategy for generation of reporter IAVs. Our strategy allows us to further expand the tolerance of IAV genome to foreign insertions. The success of the present study not only provides diverse valuable reporter IAVs and robust live-imaging mouse models of IAV infection, but also encourages generation of novel reporter viruses with more versatile capabilities, e.g., the bi- or tri- reporter viruses that express two or three foreign reporters from different segments [39]. Moreover, the feasibility of our strategy opens up better opportunities to develop live attenuated influenza virus-vectored vaccines for other highly pathogenic viruses and bacteria.

## Materials and methods

### Ethics statement

All animal experiments within this study were approved by the Institutional Animal Care and Use Committee (IACUC) of Shandong University of Traditional Chinese Medicine (Approval: SDUTCM20211230001).

### Cell culture

Human embryonic kidney cell line 293T and Madin-Darby canine kidney (MDCK) epithelial cells were grown in Dulbecco's modified Eagle's medium (DMEM; Cellgro, Manassas, VA, USA) supplemented with 10% fetal bovine serum (FBS; Gibco, Carlsbad, CA, USA), 1000 units/mL penicillin and 100 µg/mL streptomycin (Invitrogen, Carlsbad, CA, USA). Infections were performed in Opti-MEM containing 2 µg/mL N-tosyl-L-phenylalanine chloromethyl ketone (TPCK)–trypsin (Sigma-Aldrich, St. Louis, MO, USA). All cells were grown at 37 ˚C in 5% CO2.

### Plasmids

(i) Construction of the IAV minigene expressing plasmids. The plasmids expressing NS-derived reporter vRNAs were constructed by replacing the NS1-NEP ORFs with Fluc or Rluc encoding ORFs, under the control of the human RNA polymerase I (Pol-I) promoter. The $NCR_{NS}$-Fluc fragment was amplified with primers $NCR_{NS}$-Fluc-Forward/$NCR_{NS}$-Fluc-Reverse using the pISRE-Luc plasmid as template, while the $NCR_{NS}$-Rluc fragment was amplified with primers $NCR_{NS}$-Rluc-Forward/$NCR_{NS}$-Rluc-Reverse from pRL-TK plasmid. Both $NCR_{NS}$-Fluc and $NCR_{NS}$-Rluc fragment were then cloned into the *Sap* I linearized pPol-I vector using In-fusion cloning kit (Takara, Beijing, China) following the manufacturer's protocol, generating $pNCR_{NS}$-Fluc and $pNCR_{NS}$-Rluc respectively.

Next, $NCR_{NS}$-Fluc fragments carrying panhandle-stabilizing mutations were amplified using indicated primer $NCR_{NS}$-x-Forward paired with $NCR_{NS}$-Fluc-Reverse from $pNCR_{NS}$-Fluc template, and then cloned into pPol-I as described above, generating $pNCR_{NS}$-x-Fluc mutants.

(ii) Construction of plasmids expressing mutated NS-Gluc vRNAs. In order to introduce desired mutations into the modified NS-Gluc segment, NCRns-3,8m-Forward and NCRns-3,5,8m-Forward were respectively paired with the primer 5'-NCR-Reverse to amplify NS$^{CE1}$-Gluc and NS$^{CE2}$-Gluc fragments from the original pDZ-NS-Gluc plasmid [25]. The two fragments were then cloned into pPol-I vector as described above, generating pPolI-NS$^{CE1}$-Gluc and pPolI-NS$^{CE2}$-Gluc respectively. As controls, the natural NS, original NS-Gluc and natural M were amplified using primers 3'-NCR-Forward/5'-NCR-Reverse and cloned into pPol-I vector, generating pPolI-NS, pPolI-NS-Gluc, and pPolI-M, respectively.

(iii) Construction of plasmids expressing wildtype and mutated NS-Fluc vRNAs. The modified NS-Fluc was constructed using similar strategy for construction of NS-Gluc, except that an additional 2A protease sequence was inserted between NS1 and Fluc to avoid fusion. The NS-Fluc construct was divided to 3 fragments for initial amplification. The left 3'-NCR-NS1 and right NEP-5'-NCR fragment were amplified using pDZ-NS-Gluc as template with primers 3'-NCR-Forward/NS1-2A-Reverse and 2A-NEP-Forward/5'-NCR-Reverse, respectively. The middle 2A-Fluc fragment was amplified using primers 2A-Fluc-Forward/Fluc-2A-Reverse from the template pNCRns-Fluc. Adjacent fragments overlap for at least 15 nts. All the three fragments and the *Sap* I linearized pPol-I vector were then ligated using the In-fusion cloning kit, generating wildtype pPolI-NS-Fluc plasmid.

To introduce desired mutations into the modified NS-Fluc segment, NCRns-3,8m-Forward and NCRns-3,5,8m-Forward were respectively paired with the primer 5'-NCR-Reverse to amplify NS$^{CE1}$-Fluc and NS$^{CE2}$-Fluc mutant fragments from pPolI-NS-Fluc. The two fragments were then cloned into pPol-I vector as described above, generating pPolI-NS$^{CE1}$-Fluc and pPolI-NS$^{CE2}$-Fluc respectively.

All the primer sequences used above were shown in supplementary S1 Table.

## Reverse genetics

The influenza A/Puerto Rico/8/1934 virus (H1N1, PR8) and recombinant reporter viruses were generated and propagated as described previously [25]. In brief, the IAV rescue plasmids of PR8 backbone including pDZ-PA, -PB1, -PB2, -NP, -HA, -NA, -M, and -NS1, were co-transfected into 293T cells using Lipofectamine 2000 (Invitrogen, USA) according to manufacturer's instructions. At 24 hour post transfection (h.p.t.), fresh MDCK cells were seeded and co-cultured with 293T cells. After 48 h incubation, the influenza PR8 virus was harvested from the supernatant. After plaque purification, the virus was amplified in 10-day-old chicken embryos. The viral titer was determined by inoculation of serial 10-fold dilutions of stock virus onto MDCK cells and calculated by the Reed-Muench method [25].

For modified viruses, the rescue plasmid pDZ-NS1 were replaced with indicated pDZ-NS-Gluc, pPolI-NS$^{CE1}$-Gluc, pPolI-NS$^{CE2}$-Gluc, pPolI-NS-Fluc, pPolI-NS$^{CE1}$-Fluc and pPolI-NS$^{CE2}$-Fluc for construction of recombinant viruses PR8-NS-Gluc, PR8-NS$^{CE1}$-Gluc, PR8-NS$^{CE2}$-Gluc, PR8-NS-Fluc, PR8-NS$^{CE1}$-Fluc and PR8-NS$^{CE2}$-Fluc, respectively. X31-NS$^{CE2}$-Fluc was generated using rescue plasmids for the six internal segments of PR8-NS$^{CE2}$-Fluc, as well as pDZ-X31-HA and pDZ-X31-NA that encode HA and NA of X31 respectively.

## The dual-template RdRp assays

The dual-template reporter RdRp assay was conducted as previously described with slight modifications [16]. Briefly, the minigene expressing plasmids pNCR$_{NS}$-Fluc (or its mutants pNCRx-Fluc) and pNCR$_{NS}$-Rluc were co-transfected into 293T cells with IAV RdRp constituent expressing plasmids pFlu-NP, pFlu-PB1, pFlu-PB2, and pFlu-PA using Lipofectamine

2000 (Invitrogen, USA) according to manufacturer's instructions. At 24 h.p.t., the cells were harvested, and a proportion of the cells were removed for luciferase assays using Dual-Glo Luciferase Assay System (Promega, Madison, WI USA), while the left cells were extracted for total RNAs using Simply P Total RNA Extraction Kit (Bioflux, Zhejiang, China). The vRNA and mRNA were reverse transcribed using PrimeScript RT reagent Kit with gDNA Eraser (Takara, China) with the NS-specific primer RT-vRNA-NS and oligo(dT), respectively, followed by quantitative-PCR using Fluc-specific and Rluc-specific qPCR primers. The relative level of Fluc were normalized to Rluc for vRNA, mRNA as well as luciferase activity were calculated to reflect the efficacy of vRNA replication, transcription and protein expression of indicated minigene constructs.

Alternatively, the natural or modified NS segments expressing plasmids were separately co-transfected with natural M-expressing plasmid (pPolI-M) into 293T cells expressing RdRp constituents. After 24h incubation, the cells were harvested for total RNA extraction. The vRNA and mRNA were reverse transcribed as described above, except for vRNA M were reverse transcribed using a universal 3'NCR primer RT-3'NCR, followed by qPCR using NS-specific and M-specific qPCR-primers. The vRNA and mRNA level of NS-derived segments were normalized to those of M to reflect their replication and transcription efficacy respectively.

The primers used for reverse transcription and qPCR were shown in supplementary S2 Table.

### *In vitro* growth curves

MDCK cells growing in 6 well plates were infected by indicated viruses at a multiplicity of infection (MOI) of 0.01 $TCID_{50}$/cell. After 1 h incubation at 37˚C, cells were washed and fresh Opti-MEM containing 2 μg/ml TPCK-trypsin were added. Aliquots were removed at various time points for viral titration and luciferase assays.

### Luciferase assays

To determine the activity of Gluc, 50 μL of viral culture medium or lung tissue homogenate (appropriate dilution applied to avoid over range) were mixed with 50 μL of luciferase substrate using Pierce *Gaussia* Luciferase Flash Assay Kit (Thermo Scientific, Rockford, IL, USA) according to the manufacturer's instructions. The luminescence was detected immediately using Sirius L Tube Luminometer (Berthold, Germany).

Fluc assays were performed using a Britelite plus Reporter Gene Assay System (PerkinElmer, Waltham, MA, USA) according to the manufacturer's instructions. In brief, MDCK cells growing in 96 well plates were infected by indicated viruses at an MOI of 0.01. After 1 h incubation at 37˚C, cells were washed and fresh Opti-MEM containing 2 μg/ml TPCK–trypsin were added. At 24 h post infection (p.i.), the culture medium was discarded, followed by sequentially adding 50 μL PBS and 50 μL substrate. After incubation for 10min, the luminescence was detected immediately using BioTek SYNERGY neo2 Microplate Reader (BioTek, Winooski, VT, USA).

### Genome stability analysis

The indicated recombinant IAVs were serially passaged in chicken embryos for at least five passages. The viruses of each passage were tittered and used to infect MDCK cells grown in white 96-well plates (10,000 cells/well) at an MOI of 0.1. The cells were harvested at 24 h.p.i. for luciferase assays.

## Animal models

Female BALB/c mice (4 to 6 weeks old) were used in this study. All animals were maintained under specific pathogen-free conditions and all efforts were made to minimize any suffering and the number of animals.

To determine the lethality of the viruses, five mice from each group were inoculated intranasally under isoflurane anesthesia with 10-fold serial dilutions containing $10^0$ to $10^5$ $TCID_{50}$ (30 μl) of virus. Body weight and survival were monitored daily for 14 days.

To measure virus replication in mice, three to six mice in each group were inoculated intranasally under isoflurane anesthesia with indicated sublethal dose of viruses. At indicated time points, mice were subjected to *in vivo* imaging, *ex-vivo* imaging or determination of viral load/luciferase activities in lung tissues.

For antiviral treatments, 10–30 mg/kg/day of oseltamivir phosphate or vehicle only (PBS) were administered via intraperitoneal (i.p.) injection. The treatments were given twice daily for 5 days starting at 2 h before virus inoculation.

## *Ex-vivo* imaging

Mice infected with sublethal doses of PR8-NS-Gluc, PR8-NS$^{CE1}$-Gluc or mock infected were euthanized on day 3 p.i. and the trachea and lungs were excised. A syringe needle was inserted into the opening of the trachea and 0.5 ml of coelenterazine (50 μg/ml, NanoLight Technology, AZ, USA) was injected into the lung, followed by imaging immediately using IVIS200.

## *In-vivo* imaging

Mice infected with varies doses of PR8-NS$^{CE2}$-Fluc or X31-NS$^{CE2}$-Fluc were anaesthetized and the substrate D-Luciferin (PerkinElmer, Waltham, MA, USA) was injected intraperitoneally at 150 mg/kg. At 10 min after substrate administration, images were acquired with the Xenogen IVIS 200 and analyzed using the Living Image software (version 4.4).

To measure virus replication, live imaging was conducted daily, while for antiviral determination, the imaging were conducted on days 2 and 5 p.i.

## Statistical analysis

For PR8-NS$^{CE1}$-Gluc based high-throughput screening approach, the quality was assessed by evaluation of the signal-to-noise (S/N) ratio, coefficient of variation (CV) and Z' factors. (1) S/N = mean signal of negative control / mean signal of positive control; (2) CV = SD of negative control / mean of negative control; (3) Z' = $1–3 \times$ (SD of positive control + SD of negative control) / (mean of negative control—mean of positive control). SD represents the standard deviation. A Z' value between 0.5 and 1.0 is considered robust enough for an HTS assay.

Statistical significance was determined using unpaired Student's t-test with two-tailed analysis and the GraphPad Prism 5 software package (GraphPad Software). Data are considered significant when P values are <0.05.

## Supporting information

**S1 Fig. The effects of panhandle-stabilizing mutations on vRNA transcription.** The indicated NS-derived vRNAs were separately subjected to dual competition assay with wildtype M as competitive control. Data represents the relative mRNA levels of wildtype M, NS and normalized NS/M. *, p<0.05; **, p<0.01; ***, p<0.001; ns, no significance; students' t test. (TIF)

**S2 Fig. Genomic stability analysis.** (A) The indicated reporter viruses were serially passaged in chicken embryos and the titer of each passage was determined. Error bars indicated Mean ± SEM of three independent experiments. (B) The genome RNAs of indicated recombinant viruses were extracted using TIANamp Virus RNA Kit (Tiangen, China). The complementary DNA was prepared using PrimeScript RT reagent Kit with gDNA Eraser (Takara, China) and NS segment specific primer (5'-CAGGGTGACAAAGACATAATG-3'). Then PCR analysis was performed using the 2xTaq MasterMix (Cwbio, China) and primers covering the full length of firefly luciferase gene (NS-Fluc-specific-Forward:5'-ACGTCGAGGAGAATCC CGGGCCCATGGAAGACGCCAAAAA-3'; NS-Fluc-specific-Reverse: CAGGCTAAAG TTGGTCGCGCCGCTGCCCAATTTGGACTTT). The PCR product was analyzed using 1% agarose gel electrophoresis. The plasmids pPolI-NS^CE1^-Fluc and pPolI-NS^CE2^-Fluc were used as positive controls, while pPolI-NS was used as the negative control.
(TIF)

**S3 Fig. Deep sequencing analysis.** The viral RNA of PR8-NS^CE2^-Fluc (Passage 5) was extracted using SparkZol reagent (SparkJade, China) according to the manufacturers manual and sequentially subjected to first and second strand cDNA synthesis using BeyoRT II First Strand cDNA Synthesis Kit (RNase H⁻) and Second Strand cDNA Synthesis Kit (Beyotime, China). The double stranded cDNA library of virus genome was deep sequenced by Novogene (China) using Illumine nova 6000. Structural variation (SV) analysis by BreakDancer software (V1.4.4, http://breakdancer.sourceforge.net/) detected no insertion, deletion, inversion and translocation of the large segments in the genome level, While SNP/InDel analysis using SAM-TOOLS identified two substitutions but no insertion/deletion. (A) The depth of sequencing to positions along the reference genome. (B) Identification and characterization of the mutations.
(TIF)

**S1 Table. Primers for construction of the plasmids.** [a] The substitutions at indicated positions are shown in red.
(DOCX)

**S2 Table. Primers for reverse transcription and qPCR analysis.**
(DOCX)

## Acknowledgments

We thank Prof. Dongmei Qi and Dr. Xiwen Geng from Experiment Center of Shandong University of Traditional Chinese Medicine for their technical support.

## Author Contributions

**Conceptualization:** Lijun Rong, Qinghua Cui, Ruikun Du.

**Data curation:** Qinghua Cui, Ruikun Du.

**Formal analysis:** Xiujuan Zhao, Ruikun Du.

**Funding acquisition:** Qinghua Cui, Ruikun Du.

**Investigation:** Xiujuan Zhao, Xiaojing Lin, Ping Li, Zinuo Chen, Chengcheng Zhang.

**Methodology:** Xiujuan Zhao, Xiaojing Lin, Ruikun Du.

**Resources:** Xiujuan Zhao, Qinghua Cui.

**Supervision:** Lijun Rong, Ruikun Du.

**Validation:** Ping Li, Qinghua Cui.

**Writing – original draft:** Xiujuan Zhao, Ruikun Du.

**Writing – review & editing:** Balaji Manicassamy, Lijun Rong, Qinghua Cui, Ruikun Du.

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
