## [Decision Letter · Decision Letter 0]

26 Apr 2022

Dear Dr. Du,

Thank you very much for submitting your manuscript "Expanding the Tolerance of Segmented Influenza A Virus Genome Using a Balance Compensation Strategy" for consideration at PLOS Pathogens. As with all papers reviewed by the journal, your manuscript was reviewed by members of the editorial board and by several independent reviewers. In light of the reviews (below this email), we would like to invite the resubmission of a significantly-revised version that takes into account the reviewers' comments.

We cannot make any decision about publication until we have seen the revised manuscript and your response to the reviewers' comments. Your revised manuscript is also likely to be sent to reviewers for further evaluation.

Sincerely,

Daniel R. Perez, PhD

Associate Editor

PLOS Pathogens

Carolina Lopez

Section Editor

PLOS Pathogens

Kasturi Haldar

Editor-in-Chief

PLOS Pathogens

orcid.org/0000-0001-5065-158X

Michael Malim

Editor-in-Chief

PLOS Pathogens

orcid.org/0000-0002-7699-2064

Reviewer's Responses to Questions

**Part I - Summary**

Reviewer #1: The work by Zhao and collaborators (Expanding the Tolerance of Segmented Influenza A 1 Virus Genome Using a Balance Compensation Strategy) explores the gene expression of IAV after introducing foreign genes in the NS segment using a widely described laboratory-adapted strain such as PR8. The introduction of exogenous genes such as reporter genes into IAV viruses is of crucial relevance to the research community to help in the understanding of different viral processes and the evaluation of countermeasures against Influenza. Unfortunately, one of the main problems regarding the generation of recombinant viruses with heterologous sequences is the modifications of viral properties such as replication and virulence in vivo. The authors tackle this problem by the introduction of mutations in the NCR regions of the NS segments showing a modulation in the vRNA, mRNA, and protein synthesis, followed by characterization in vitro and in vivo of the replication capacity, virulence, and reporter expression; the main conclusion obtained is that a triple mutation of the 3’NCR or NS allows for the introduction of a FLuc reporter which is stable based on Fluc activity, exhibiting similar growth profiles in vitro and 100% lethality after a high dose challenge in vivo.

Overall, the hypothesis and the strategy are clear. The experiments proposed are in line with the answers that the authors should provide. However, one of the main flaws in the work is the segment and strain dependency of the phenomenon observed. It is well established that PR8 is a highly laboratory-adapted strain with an increased tolerance for genome rearrangements and attenuation strategies. Based on this, the relevance of the work is compromised since we cannot establish the applicability of this type of rearrangement in more recent strains such as seasonal H3N2 or pandemic H1N1. For instance, the use of this strategy for the assessment of antiviral efficacy should be performed against current circulating strains, so the relevance of the results presented is conditioned to the tolerance of the rearrangement by different strains and not just PR8.

Furthermore, is this strategy capable of tolerating different HA/NA subtypes? How is the plasticity of the rearrangement studied when other HA/NA combinations of relevance are incorporated? This is in line with the applicability and significance of the strategy that the authors did not address. This reviewer’s opinion is that besides the experimental concerns shown below, the relevance and importance of this strategy need to be studied further. All the mentioned above could improve the originality of the work.

Reviewer #2: In the manuscript presented by Zhao et al, the authors researched on the improvement of influenza A segment incorporation when segments are modified for the incorporation of exogenous genes. This is very important for the field because there is an increasing interest for the use of modified influenza A viruses. One major caveat is that these viruses generally exhibit attenuation. The authors have previously demonstrated that one of the major issues relates to an imbalance in the generation of the gene segments. Basically, the modified segment produces less vRNA which in turns leads to attenuation. To overcome this problem, the authors improved the incorporation by mutagenizing the none coding regions (NCR) of a modified (PR8) segment encoding for either firefly or renilla luciferase. They used an in vitro competitive assay in which the WT NCR and the mutant(s) compete. They showed that the mutants were more efficient at producing the reporter, and vRNA and mRNA from the modified segment. However, the in vivo (mice) experiment showed that the improvement was partial; the replication of one of the mutant viruses reached titers comparable to wt but the pathogenesis were not comparable. The mutant virus did not show neither weight loss nor the lethality of wt PR8. The improvement had a more significant impact when the reporter used was larger. The manuscript needs some improvement in terms of writing. Even though the English is clear, there are some colloquial expression that must be avoided. Also, the discussion needs more work. The manuscript presented can be considered for publication after major revisions.

**Part II – Major Issues: Key Experiments Required for Acceptance**

Reviewer #1: Major comments

Fig 1A shows an empty table. I am not sure if this is an error in the PDF version or the version submitted by the authors. In addition, the scale used in 1C does not allow the view of nuanced differences among groups.

In vitro kinetics (Fig2): It is highly speculative to state an improved replication (‘álmost identical’’) of 38m-Gluc virus compared to PR8-NS Gluc and similar replication when compared with PR8 wt. Results presented are skewed by sharp differences observed at 12h, suggesting differences at time 0, which cannot be ruled out based on the information given. Furthermore, between times 12 and 36, you see (roughly) a three log10 fold increase for PR8 wt, a two-fold increase for NS-Gluc, and just a 1.5 fold increase for the mutants, so I disagree with the authors about the conclusions raised.

What rationale was employed to exclude NS358m from the in vivo experiments in figure 3? As I stated previously, in vitro growth and kinetic data need to be carefully examined. On the other side, NS358m has the higher replication level shown in 2B.

Although the authors provided stability data analyzed using FLuc activity, there is no sequence data about the NS segment and the rest of the IAV genome. What are the nucleotide sequence in the NS segment and the other segments? Does any new nucleotide change appear? Are the three introduced mutations maintained? How does the inclusion of Fluc affect the stability of the HA, for instance, in comparison with the wt virus? All these questions fundamental to understanding the stability of the virus generated remain unanswered. In addition, no information about yield is provided regarding the recombinant viruses. Is the yield comparable to the wt virus?

How many MDL50 related to the PR8 wt is the high dose of PR8-NS358m-Fluc used?

Reviewer #2: Even though some of the findings seems to address the problem, I think the authors could have dug deeper. In my view, the number of sites and virus used fall short. The authors need to increase the number of viruses and mutation to complement, at least, the in vitro part. I would also try to use other gene segments.

One major problem with the production of enlarging influenza A segments is that these are usually prone to an increment in the production of defective interfering (DI) particles. The authors did not mention this possibility. I would measure DIs to see if the mutants are less prone to the production of DI particles. This could actually be the reason for the incomplete phenotype showed by the mutants when compared to WT.

Where the mutant viruses stable? You showed in figure 4C that the PR8-NS358m-Fluc maintained the phenotype, but you do not show any sequencing. The authors need to sequence all the mutant viruses to show stability from the in vitro and in vivo experiments.

**Part III – Minor Issues: Editorial and Data Presentation Modifications**

Reviewer #1: Were samples treated with DNAase for in vitro assays?

Line 140: Please explain briefly how mutations were introduced

Line 392: typo (sepcific)

Fig 2B: Please clarify the nomenclature used for each group and their relationship with the different mutations. The differences in terminology in the text and figure are confusing.

Fig 2D: TCID should say TCID50 in the title

Fig 3C: Baloxivir needs to be corrected

Reviewer #2: Figure 1. The table looks empty on my end.

How well conserved are these alleles in the databases

Figure 2B. Please change nomenclature in the figure from NS358m to something like NS3,5,8m or an alternative that helps on differentiating these mutations to mutations on nucleotide position 358.

Supplementary figure 1. Change y-axis legend from mRNA to vRNA

Line 169-170. “Over-compensation would on the other hand disrupt the balance of the multiple segments, which also results in attenuation”. You are assuming this is the main mechanism by which NS358m-Gluc is less efficient in vivo. But it can be any other, like cells being more sensitive to RNA detection, stronger IFN response, etc. Which at the end of the day they are all related but you guys do not show any on this respect. Since this is still speculation, I suggest to either remove the interpretation or move it to discussion and address as speculative.

Line 209 live imaging remains a mission impossible. There is no impossible for science. We just don’t know yet.

As Fig 4B shows, the replication of the original NS-Fluc is impaired much worse compared to NS- Gluc, not to mention the natural NS. I think there could be more proper ways to write about the results.

Line 275. Please, find a different word for “Encouragingly”.

Line 305-306. “…may contribute a lot”. Please, reword.

PLOS authors have the option to publish the peer review history of their article (what does this mean?). If published, this will include your full peer review and any attached files.

Reviewer #1: No

Reviewer #2: No
---

## [Editor Report · Decision Letter 1]

21 Jul 2022

Dear Dr. Du,

We are pleased to inform you that your manuscript 'Expanding the Tolerance of Segmented Influenza A Virus Genome Using a Balance Compensation Strategy' has been provisionally accepted for publication in PLOS Pathogens.

Best regards,

Daniel R. Perez, PhD

Associate Editor

PLOS Pathogens

Carolina Lopez

Section Editor

PLOS Pathogens

Kasturi Haldar

Editor-in-Chief

PLOS Pathogens

orcid.org/0000-0001-5065-158X

Michael Malim

Editor-in-Chief

PLOS Pathogens

orcid.org/0000-0002-7699-2064
---

## [Editor Report · Acceptance letter]

2 Aug 2022

Dear Dr. Du,

We are delighted to inform you that your manuscript, "Expanding the Tolerance of Segmented Influenza A Virus Genome Using a Balance Compensation Strategy," has been formally accepted for publication in PLOS Pathogens.

Best regards,

Kasturi Haldar

Editor-in-Chief

PLOS Pathogens

orcid.org/0000-0001-5065-158X

Michael Malim

Editor-in-Chief

PLOS Pathogens

orcid.org/0000-0002-7699-2064